# Should Undergraduate Lectures be Compulsory? The Views of Dental and Medical Students from a UK University

**DOI:** 10.3390/dj5020015

**Published:** 2017-03-31

**Authors:** Alaa Daud, Aaron Bagria, Kushal Shah, James Puryer

**Affiliations:** School of Oral & Dental Sciences, Bristol Dental Hospital, Lower Maudlin Street, Bristol BS1 2LY, UK; ab12165@my.bristol.ac.uk (A.B.); ks12273@my.bristol.ac.uk (K.S.); james.puryer@bristol.ac.uk (J.P.)

**Keywords:** undergraduate, lecture, compulsory

## Abstract

Formal lectures have been a traditional part of medical and dental education, but there is debate as to their compulsory status. This study was designed to explore dental and medical students’ views on compulsory lectures and the use of Video-Recorded Lectures (VRL). A cross-sectional study of University of Bristol students in Years 2 to 4 was conducted using an online questionnaire. The majority of both dental (76%) and medical (66%) students felt lectures should be non-compulsory. The most common learning resources used by both dental and medical students were live lectures, lecture handouts and VRL. The majority of both dental (84%) and medical (88%) students used VRL. Most students attended lectures all of the time both before and after the introduction of VRL, even though most dental and medical students believe lectures should be non-compulsory. VRL is a popular learning resource. These findings tie-in with General Dental Council and General Medical Council recommendations that encourage self-directed learning. Dental and Medical schools should offer a range of learning resources and make use of current technology, including the use of VRL.

## 1. Introduction

Traditionally, formal large-group lectures have been an important pedagogical component of medical and dental education [1,2], although recent studies have shown that there is now debate as to whether or not attendance at lectures should be compulsory [3,4,5].

Universities have also been promoting the search for new learning resources to enhance the student learning experience [6], and one resource progressively favored by students is the use of video-recorded lectures (VRL) [7]. Dental and Medical schools within the UK have begun to incorporate VRL into their taught undergraduate courses and the advantages and disadvantages of VRL have been widely discussed elsewhere [8,9,10,11,12,13,14,15,16,17,18,19,20,21,22,23,24]. Of the universities currently offering VRL, some deem lecture attendance non-compulsory, and for students at these universities, VRL gives them the option of either attending live lectures, utilising VRL, or both. In contrast, some universities adopt the opposite stance deeming lecture attendance compulsory, despite the introduction of VRL. This is due to concerns regarding falling lecture attendance levels [25], relying on video-recording technology that may not work [14] and concerns that students will miss out on lecturer’s non-verbal language including bodily gestures, which can help emphasise information in a particular way [26].

In September 2015, the University of Bristol (UoB) introduced VRL through Mediasite, a video-recording technology that records both the audio and visual content of a lecture. The Dental School was an ‘early adopter’ of the scheme and took part in the piloting process. Currently, all undergraduate lectures within the Dental School use VRL. Lectures are typically 40–50 min in length, and are usually scheduled at either 8 a.m. and/or 9 a.m. each weekday. Following the lecture, a draft version of the VRL is sent to the relevant lecturer who has the opportunity to review and edit the presentation before it is placed online for students to view. Students can normally access the VRL through the School’s online portal within 48 h of the live lecture.

Lecture attendance within the Dental and Medical Schools is still currently compulsory. Within the Dental School, students are required to complete a ‘sign-in’ sheet for each timetabled lecture, and this information is stored with the School administration team. At the end of each term, lecture attendance (along with other data such as clinical grades and absenteeism) levels are reviewed for each student at a Progress Meeting. Students who are found to have missed seven or more lectures that term are required to have an interview with their Personal Tutor and those students who have missed 12 or more lectures are normally interviewed by the Clinical Dean or Head of School. Ultimately, persistent offenders may be referred to the Faculty ‘Fitness to Practice’ panel.

Following the introduction of VRL, there is now a growing consensus within the student body at the University of Bristol that lectures attendance should not be compulsory.

## 2. Aims

The aim of this study was to explore the views of undergraduate dental and medical students at the University of Bristol regarding the use of VRL, and whether or not lectures should be compulsory.

## 3. Materials and Methods

A cross-sectional survey of all dental (*n* = 202) and medical (*n* = 680) undergraduates in Years 2–4 studying at the University of Bristol was carried out. Year-1 students were not asked to participate as VRL had been introduced before they commenced study. Similarly, Year-5 students were not asked to participate as they had received very few timetabled live lectures since VRL was introduced. Thus neither Year-1 nor Year-5 students would be able to make valid comparisons. An online questionnaire was developed (Appendix A) that was designed to explore student views on the use of VRL, their reasons for attending or not attending live lectures and whether or not lectures should be compulsory. The questionnaire consisted of five sections: learning resources currently used by students, views before the introduction of VRL, views following the introduction of VRL, views on the compulsory status of lectures and finally demographic information. A mix of question styles was used combining both ‘tick-box’ responses and ‘free-response’ questions. Some questions allowed more than one option to be selected.

An introductory e-mail was distributed to the students along with a Participant Information Sheet which explained the nature of the survey and the anonymity of responses. The e-mail also contained a link to the online questionnaire (www.surveymonkey.com). Participation was non-compulsory and it was assumed that consent to participate was given by the voluntary completion of the questionnaire. Follow-up e-mails were sent to all students in Years 2–4 two weeks and four weeks following the initial e-mail invitation to participate.

SPSS software (version 23, IBM, New York, NY, USA) was used to analyze the results.

Full ethical approval from the Faculty of Medicine and Dentistry Ethics Committee (Bristol, UK) was obtained prior to the study (FREC No. 32142).

## 4. Results

### 4.1. Demographics

There were responses from *n* = 89 dental undergraduates and *n* = 133 medical undergraduates giving an overall response rate of 25.1%. The majority of both dental (71%) and medical (67%) students were female. The demographics of respondents is shown in Table 1.

### 4.2. Learning Resources

The most popular learning resources used by dental and medical students whilst studying for examinations were the use of printed lecture handouts (89% dental, 90% medical), viewing VRL (84% dental, 88% medical) and attending live lectures (84% dental, 85% medical) (Figure 1).

### 4.3. Should Lectures Be Compulsory?

Only a minority both dental (24%) and medical (34%) students thought that lectures should be compulsory.

### 4.4. Lecture Attendance before/after the Introduction of VRL

Prior to the introduction of VRL, 52% of dental students attended lectures ‘all of the time’, 42% ‘most of the time’ and 6% some of the time (Figure 2). No students replied that they ‘rarely’ or ‘never attended’ lectures prior to the introduction of VRL. Following the introduction of VRL, there was no obvious difference in the lecture attendance pattern of dental students with 49% reporting that they attended ‘all the time’, 44% ‘most of the time, and 7% ‘some of the time’. Again, no student replied that they ‘rarely’ or ‘never’ attended lectures.

The responses from medical students showed that they had a similar pattern of lecture attendance to dental students prior to the introduction of VRL with 48% replying that they attended ‘all of the time’, 42% ’most of the time’, and 10% ’some of the time’. Again, no medical student replied that they ‘rarely’ or ‘never’ attended (Figure 3). However, following the introduction of VRL, 44% replied ‘all the time’, 42% ‘most of the time’, 8% ‘some of the time’ and 6% ‘rarely’. No medical students replied that they ‘never’ attended. This shows a marked increase in the number of medical students who attended only ‘some of the time’ following the introduction of VRL.

### 4.5. Reasons for Attending Lectures

The majority (82%) of dental students reported that they attended lectures as attendance was compulsory, whilst 62% attended ‘out of routine’ and 21% for ‘social interaction’ (Figure 4). Similarly, the majority (72%) medical students responded that they attended lectures due to their compulsory nature, whilst 53% attended ‘out of routine’ and 24% for ‘social interaction’.

### 4.6. Reasons for Missing Lectures before/after the Introduction of VRL

Prior to the introduction of VRL, the most common reasons cited by dental students for not attending lectures were the quality of lectures (69%), the early start time of the lecture (55%) and medical reasons (44%) (Figure 5). Following the introduction of VRL, the most common reasons cited for missing lectures were the early start time of the lecture (58%), the lecture quality (45%) and medical reasons (41%).

Similar reasons for missing lectures prior to the introduction of VRL were given by medical students (Figure 6). A large number (69%) missed lectures due to poor quality of teaching, an early start time of the lecture (54%) and changes in the timetable (44%). Following the introduction of VRL, similar reasons were given for missing lectures with 67% citing poor lecture teaching, the early lecture start time (51%) and changes to the timetable (40%). Only 4% reported that they did not attend lectures due to availability of VRL.

### 4.7. Reasons for Student Use/Non-Use of VRL

The most frequent responses given by dental students as to why they use VRL were ‘watch at any time of the day’ (90%), ‘pause to look up information’ (85%) and ‘exam preparation’ (84%) (Figure 7). The most frequent responses for reasons that dental students do not use VRL were ‘lack of motivation’ (53%), ‘lack of time’ (48%) and ‘already obtained the information from attending the lecture’ (39%) (Figure 8).

Medical students responded with similar reasons for using Video-Recorded Lectures (VRL), the most frequent being ‘pause to look up information’ (93%), ‘watch at any time of the day’ (89%) and ‘exam preparation’ (88%) (Figure 7). Reasons for not using VRL were again similar to those for dental students with medical students responding ‘lack of time’ (62%), ‘already obtained the information from attending the lecture’ (49%) and ‘lack of motivation’ (46%) (Figure 8).

## 5. Discussion

This study found that the majority of both dental and medical students believe lectures should be non-compulsory which supports findings of previous studies [27,28]. A number of students stated that it should be the student’s own responsibility to decide whether or not they need to attend live lectures in order to gain sufficient knowledge to be successful in exams, that live lectures did not suit their learning style and they particularly disliked the early morning scheduling of live lectures. Examples of qualitative comments from the free-text responses included “*found it easier to learn and understand the content when going over lectures in own time as opposed to a morning lecture*”, “*attending 8 a.m. lectures physically is of no benefit as difficult to concentrate and make notes*”, “*I am dyslexic and I am forced into this learning style that does not suit me whatsoever*”, “*I turn up to achieve attendance only and gain nothing from it*” and “*certainly during exam times lectures should not be compulsory- students stay up late revising and they are then expected to turn up for an 8 a.m. and 9 a.m. lecture, and then treat patients 10 a.m. until 5 p.m.!*”. However, some educators disagree with this student view that lectures should be non-compulsory. They believe students should not perceive attending lectures solely as a route to passing exams, and that they offer clinical students additional benefits including improving their clinical performance in aspects such as taking a patient medical history [29], providing a shared learning experience, allowing students to ask questions for clarification, allowing information to be shared with all students and allowing a lecturer’s body language to add emphasis [30].

The current study found that both dental and medical students use a variety of learning resources for studying and exam preparation. The high number of dental (84%) and medical (88%) students using VRL suggests that they are a very popular resource and supports the findings of two previous studies [31,32]. A similar number of both dental and medical students used VRL, live lectures and lecture handouts, suggesting that students used a blended range of learning resources for learning, again supporting previous findings [8]. One interesting finding from our study is that many more medical students (67%) compared to dental students (16%) watch the VRL at ‘double speed’. In addition, almost twice as many medical students than dental students use practice exam questions to help their study. The reasons for both of these findings are unclear and would warrant further investigation.

The introduction of VRL did not have any significant effect on the number of both dental and medical students who attended lectures ‘all of the time’, ‘most of the time’ and ‘some of the time’ which suggests that the use of VRL does not affect lecture attendance. This finding may allay educators concerns that the use of VRL may increase student absenteeism from live lectures [33]. A lack of significant correlation between the introduction of VRL and declining lecture attendance levels have been widely reported [34,35,36]. Whilst one earlier study reported reduced attendance levels following the introduction of VRL [37], this may not have been solely due to the introduction of VRL. Additional factors that contribute towards reduced lecture attendance have included student health issues and the inconvenient schedule time of the lectures [38]. These findings are supported by our own study where students gave similar reasons for missing lectures following the introduction of VRL. In addition, our own study found that the two most common reasons for both dental and medical students to miss lectures was the poor quality of lectures and the inconvenient lecture times. However, at the current time lectures are still compulsory which means that even if students found the use of VRL improved their learning compared to live lectures, they still felt compelled to attend the live lectures.

The reported poor quality of lectures by students could be researched further to determine which aspects need improving. Lecture quality is important and is not enhanced with technology—a recorded poor lecture is still a poor lecture. The scheduling of lectures is a problem faced by many dental and medical schools. Due to factors such as the intensive nature of the curriculum, complex timetabling whereby student Year-groups are often split into smaller teaching groups (often over multiple locations) and the need to fit in with patient clinics, it is often difficult to schedule lectures where all students within a Year-group are available. Often this means that lectures will be scheduled at 8 a.m. before patient clinics begin, a time that is usually unpopular with students, as found in our study. The alternative of scheduling them at the end of the day is also problematic as patient clinics may overrun causing students to be late and student learning may be reduced as students may have already been studying or treating patients for many hours.

The most common reason (93%) that medical students gave for using VRL is so that they can pause and look up information, supporting an earlier study [39]. The most common reason (90%) that dental students gave for using VRL was so that they could view them at any time of the day. Again, this supports earlier findings [8]. Other benefits of using VRL have been previously reported including reducing course related student anxiety [40] and improving confidence in exams [41]. As the reasons for using VRL were equally favored by dental and medical students in our own study, it suggests that the benefits of VRL are multifactorial and this has previously been reported [42]. However, some students did not perceive VRL to be of value and the most common reason given by dental students (53%) for not using them was ‘lack of motivation’ and by medical students (62%) was ‘lack of time’. Similar student reasons have been reported previously [9]. This shows that whilst the use of VRL may be useful for some students, for others it may not, highlighting the complex nature of student learning.

This study does have some limitations. The overall response rate of 25.1% means there may be some selection bias in respondents, such that the results of the study may not be generalizable to all students. The results of the study may have been affected by recall bias, as some of the questions relied upon student recall of their attendance levels at lectures prior to, and after, the introduction of VRL. It would have been methodologically better to have used actual recorded attendances at lectures, and this would have given more reliable data. Furthermore, bias may have been introduced by the fact that the use of compulsory lectures is a somewhat contentious issue. Those with strong views on the subject may have been more inclined to respond compared to those with less strong views. Another factor that may have influenced results is that attendance at lectures still remains compulsory at the present time following the introduction of VRL.

Despite these limitations, we feel that this study adds to the current limited knowledge base and provides a contemporary benchmark from which further research could be conducted. The use of focus groups could help to explore in more detail student views on compulsory lecture attendance. It would be beneficial to repeat the study if it were decided that lectures should become non-compulsory in order to see if there was an adverse effect on live lecture attendance and especially upon student performance. Previous observational studies have found that medical students who frequently attended class have greater exam performance, course pass rate and cumulative grade averages [43,44,45], although later studies have shown that computer-based teaching yields similar knowledge gains as lecture-based teaching [46,47]. There is also an argument that attending lectures helps professional development [48]. It would also be very sensible to explore the views of teaching staff, as well as those of the students so that a more balanced view could be obtained. The Dental School is currently undertaking a Curriculum Review and the results of this study will help to determine whether or not the current compulsory status of lectures will remain.

## 6. Conclusions

This study found that the majority of both dental and medical students believed that lectures should be non-compulsory. VRL is a popular learning resource which helps to fulfil both General Dental Council (GDC) and General Medical Council (GMC) recommendations that students are offered a range of learning resources, especially those that make use of new technology [49,50]. Both the GDC and GMC also encourage self-directed learning which will help graduates maintain their knowledge and clinical skills throughout their practicing career. If lectures were to be made non-compulsory, as is the wish of the majority of students, self-directed learning may be encouraged. Further research should be carried out to determine student views on the use of VRL if a decision is made to make lectures non-compulsory, and also to see whether or not there is any adverse effect on student learning and performance.

## Figures and Tables

**Figure 1 dentistry-05-00015-f001:**
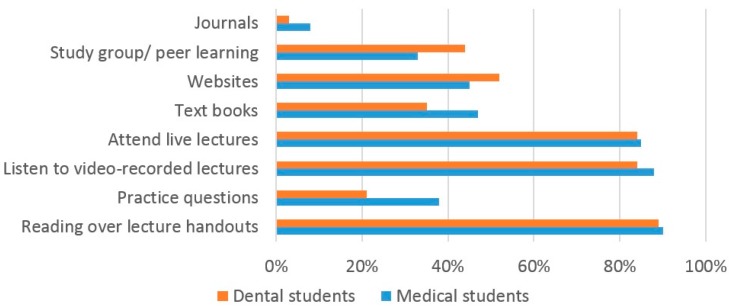
The methods used by dental and medical students to prepare for an examination.

**Figure 2 dentistry-05-00015-f002:**
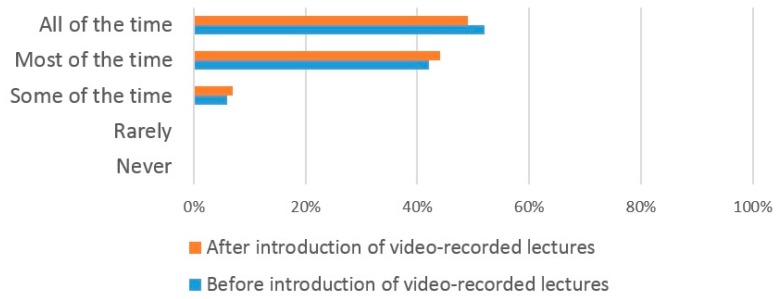
Dental student lecture attendance patterns before/after the introduction of VRL.

**Figure 3 dentistry-05-00015-f003:**
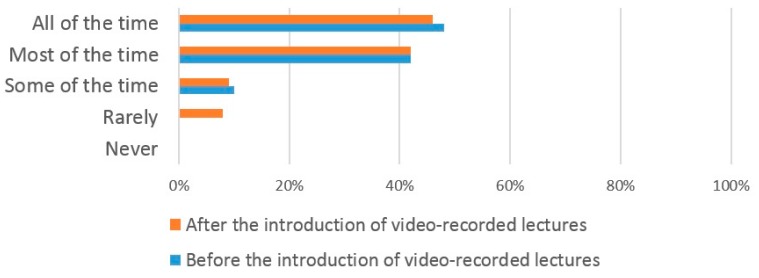
Medical student lecture self-reported attendance patterns before/after the introduction of VRL.

**Figure 4 dentistry-05-00015-f004:**
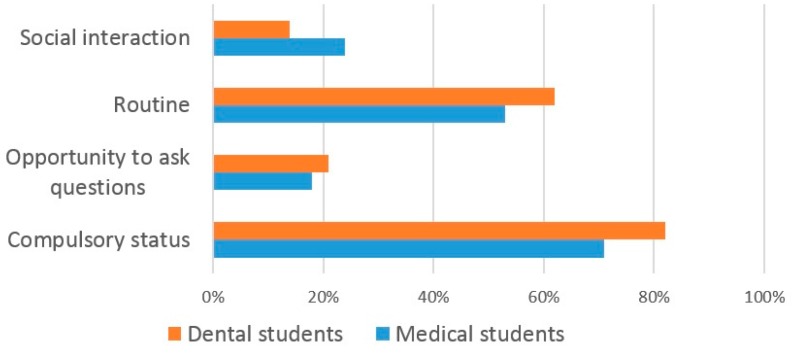
The reasons why dental and medical students attend lectures.

**Figure 5 dentistry-05-00015-f005:**
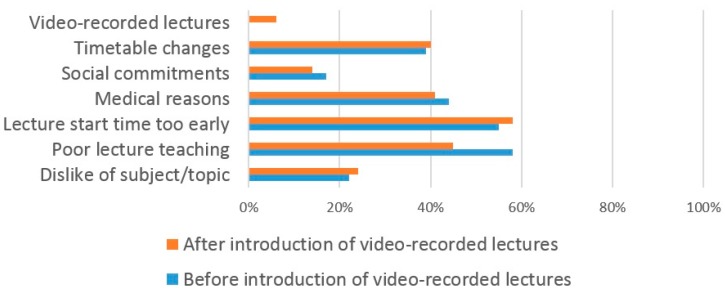
The reasons why dental students miss lectures.

**Figure 6 dentistry-05-00015-f006:**
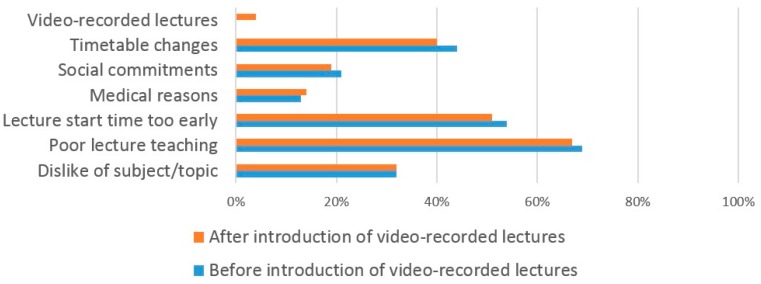
The reasons why medical students miss lectures.

**Figure 7 dentistry-05-00015-f007:**
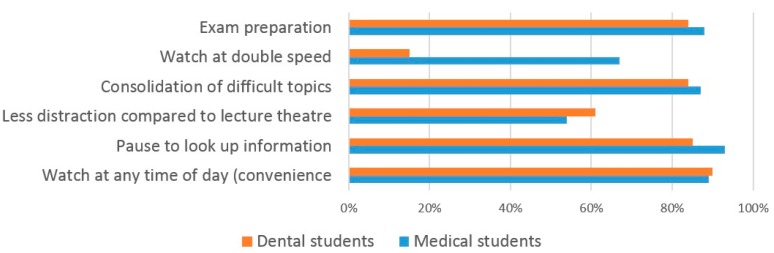
The reasons why students use VRL.

**Figure 8 dentistry-05-00015-f008:**
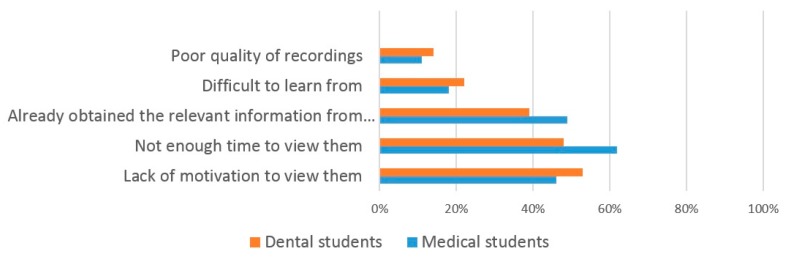
The reasons why students do not use VRL.

**Table 1 dentistry-05-00015-t001:** Demographics of respondents.

Demographic Information	Dental Students (*n*)	Medical Students (*n*)
Total	89	133
Gender	-	-
Male	26	44
Female	63	89
Year of Study	-	-
Year-2	21	52
Year-3	24	44
Year-4	34	37

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
