# Peer review of "Should Undergraduate Lectures be Compulsory? The Views of Dental and Medical Students from a UK University"

_dentistry, 2017, doi:10.3390/dj5020015_

Round 1
Reviewer 1 Report
I have made some comments in the PDF so please see there.
ON the key issues I will mention these now. there needs to be a description of the nature of the VRLs, how many, who are they by, how long are they, how are they accessed. This is critical to gain understanding. This must be corrected.
Also, it must be remembered:
"a bad lecture recorded is still a bad lecture" technology does not enhance.
The students reported on this. Was comment made that lectures should be improved?
If you were to suggest changes I would look at T and L enhancments to improve lectures.
What is a lecture in this conext? 45 minutes of sage on the stage talking the students to sleep? Especially at lunch times! I know what it is like. This is a bad and outdated model. How does the flipped classroom or small group activities impact students perceptions of lagre group learning ?
This paper tends to presume all lectures are the same which they are not or should not be.
There is no mention of questionnaire design. This is an omission. And must be corrected in future studies.
What about focus group analysis?
I think this will be of interest to readers but there is great scope to take this further. I would suggest input from medical educators with education research methods backgrounds. It takes more time but is worth it.
As a friendly reminder this type of paper is at the low end of rigor with regards to educational research methodlogy and you may want to consider AMEE online courses. I wish I had done these a decade ago!

Author Response
Thank you very much for your review of our paper and suggestions as to how it could be improved.
We have fully revised the paper based upon your suggestions. Specifically we have:
1) Described the nature of the VRLs and how they are accessed
2) Commented that "a bad lecture is still a bad lecture"
3) Discussed that views on lecture quality should be researched further
4) Described the format of lectures
5) Added the questionnaire used in the study as an Appendix
6) Suggested that focus groups could be used as a further research method
We hope that you will see view these changes in a positive light when giving further consideration to our paper.
Reviewer 2 Report
The manuscript by Daud et al describes the introduction of video recorded lecture system at the University of Bristol medical and dental school. It contains a survey of student opinions to this change.
The topic of live versus video recorded lecture has an extensive bibliographic footprint and this manuscript does not add anything new to this topic. Actually it is not a scientific study, but rather an incomplete review of a sliver of the existing literature and a report of the raw data of the survey carried out by the authors. There is no scientific/statistical analysis of these raw data and the analysis does not go beyond a confirmation of some selected publications to the topic.
However, there is one aspect to the work described in this manuscript that might form the basis of a paper that would be interest to the medical/dental education community. As the authors point out, there are very little published studies that look at mandatory versus voluntary lecture attendance. Unfortunately, the authors did not perform a comprehensive literature search and the few publications pertinent to this topic are not cited in the manuscript. Here are just a few papers addressing this question:
Zazulia and Goldhoff 2014 Teach. and Learn. in Medicine
Horvath et al 2013 Journal of Dental Education
Rysavy et al 2015 Med Sci Educ
St. Clair 1999 Innov. Higher Educ.
and other publications
Most of the manuscript deals with students’ opinion about the video-recorded lecture system. This topic has been analyzed in much detail previously, including its influence on lecture attendance and examination scores. The authors may want to focus on the mandatory versus voluntary lecture attendance policy at the University of Bristol and how this policy affects the use of live lecture attendance, VRL usage, students use of other learning resources, and potentially their learning success. First, they should conduct a thorough scientific analysis and not just report the raw data of their survey. This analysis could make use of the quantitative, as well as the qualitative data, which they already have collected. Obviously, this would need a very significant revision of the manuscript. However, in its present from, the manuscript is not reporting anything new or anything of interest to the medical/dental education community. VRL has been introduced by hundreds, if not thousands of other schools and plenty of students’ opinion papers have already been published. I would encourage the authors to redesign and refocus their manuscript and carry out a scientific analysis of their data to address the question of mandatory versus voluntary lecture attendance and how the VRL influences the students’ decision to attend class (or not).
Here are some more specific comments concerning the current manuscript:
The introduction reads like a review about the pros and cons of video-recorded lectures, a question that has been discussed extensively in the existing literature. The authors should shorten and streamline the introduction and focus on the problem of mandatory versus voluntary lecture attendance. Also the discussion needs to be refocused and shortened.
The overall response rate (25.1%) is very low. The authors acknowledge this as a limitation. However, the risk of sampling bias is not alleviated by the number of respondents (222). It doesn’t work that way.
The authors claim that the introduction of VRL did not affect lecture attendance (lines 216-8). They base this statement on only one post-VRL introduction time point. In addition, the mandatory lecture attendance policy a t the University of Bristol appears to have a significant role in this student behavior (the authors’ own resulting in Fig 4 and lines 145-6 and 240). A multi-year longitudinal study and a consideration of the mandatory attendance policy would be needed to validate this claim.
That lecture attendance is mandatory at the University of Bristol is only mentioned very later (the Material and Method section is insufficient and too short, as the introduction and the discussion sections are too long) on line 145. This raises the interesting questions whether and how this policy is enforced and whether students face any disciplinary or academic consequences when not attending lectures. That may be a good starting point for an interesting publication.
For Figure 3 it should be clarified that the data are self-reported and not actual student behavior. This is a limitation of the study. Is there any evidence in form of actual attendance records available that would support the self-reported numbers?
Are any of the differences between dental and medical students statistically significant? Do medical and dental students’ opinions and behavior differ in any way? Johnson et al (Eur J Dent Educ 2014) found some interesting differences in students’ behavior at a different university. Some appear to correlate with voluntary versus mandatory lecture attendance. Based on their own data, do the authors find similar or other differences between the two groups?
Author Response
Thank you very much for your review of our paper and suggestions as to how it could be improved.
We have fully revised the paper based upon your suggestions. Specifically we have:
1) Revised the theme of the paper to look more specifically as to whether or not lectures should be compulsory.
2) Changed the title of the paper to reflect this major change
3) Shortened the Introduction
4) Refocused the Introduction to look further at compulsory vs non-compulsory lectures
5) Added further suitable references as suggested
6) The Discussion section has been fully revised also
7) Removed the sentence related to sampling bias being compensated for by having a large number of respondents
8) Explained the mandatory attendance policy and the repercussions of non-attendance at lectures
9) Explained earlier in the paper that lecture attendance is compulsory at Bristol
10) Clarified Figure 3 to say "self reported"
11) Revised the Abstract and Conclusion
12) Added examples of the qualitative free-text responses
We hope that you will see view these changes in a positive light when giving further consideration to our paper.
Round 2
Reviewer 2 Report
This reviewer is pleased to see that the authors took the advice from the initial review and refocused the manuscript, now addressing the question of compulsory or voluntary lecture attendance. I am a bit disappointed that the authors did not take this manuscript to the next level and made it a real scientific analysis. In its present form, it is not more than a case report. Therefore, I would give it a low priority for publication in the Dentistry Journal. Before I can recommend it for publication, several small things need to be changed.
The title needs to be changed. The content of the manuscript does not answer the question posed in the present title. The data only say the students at the University of Bristol prefer non-mandatory lecture attendance.
The authors changed the manuscript about whether the introduction of the VRL changes lecture attendance. However, there still remains a sentence in the abstract stating “VRL does not affect lecture attendance”. As both dental and medical students at the University of Bristol are required to attend lectures, this statement is unsupported as the mandatory attendance policy is an overriding factor. In addition, it would require a longitudinal dataset to support such a conclusion. This manuscript is based on only one time point.
The statement that medical students are also expected to attend lectures is mentioned too late, on line 132. It should be mentioned together with the mandatory lecture attendance for dental students. In view of this study, both student populations are very similar, at least when it comes to mandatory lecture attendance. The published literature supports the notion that they would significantly differ, if one group of students had a mandatory attendance policy, whereas the other had not.
The text in the figures needs to be enlarged. As currently reproduced, the font size is much small than the body of the text, making it very cumbersome to read the text in the figures.
Author Response
Please see attached Word document

Round 3
Reviewer 2 Report
Thank you for implementing this reviewer's suggestions.